# The Dynamic Changes of Alternative Electron Flows upon Transition from Low to High Light in the Fern *Cyrtomium fortune* and the Gymnosperm *Nageia nagi*

**DOI:** 10.3390/cells11172768

**Published:** 2022-09-05

**Authors:** Jun-Bin Cheng, Shi-Bao Zhang, Jin-Song Wu, Wei Huang

**Affiliations:** 1Kunming Institute of Botany, Chinese Academy of Sciences, Kunming 650201, China; 2University of Chinese Academy of Sciences, Beijing 100049, China

**Keywords:** energy balancing, ferns, gymnosperms, photoprotection, photosynthesis

## Abstract

In photosynthetic organisms except angiosperms, an alternative electron sink that is mediated by flavodiiron proteins (FLVs) plays the major role in preventing PSI photoinhibition while cyclic electron flow (CEF) is also essential for normal growth under fluctuating light. However, the dynamic changes of FLVs and CEF has not yet been well clarified. In this study, we measured the P700 signal, chlorophyll fluorescence, and electrochromic shift spectra in the fern *Cyrtomium fortune* and the gymnosperm *Nageia nagi*. We found that both species could not build up a sufficient proton gradient (∆pH) within the first 30 s after light abruptly increased. During this period, FLVs-dependent alternative electron flow was functional to avoid PSI over-reduction. This functional time of FLVs was much longer than previously thought. By comparison, CEF was highly activated within the first 10 s after transition from low to high light, which favored energy balancing rather than the regulation of a PSI redox state. When FLVs were inactivated during steady-state photosynthesis, CEF was re-activated to favor photoprotection and to sustain photosynthesis. These results provide new insight into how FLVs and CEF interact to regulate photosynthesis in non-angiosperms.

## 1. Introduction

Under natural conditions, light intensity that is exposed to leaves changes dynamically owing to wind, cloud, and shading from upper leaves or neighboring plants; such light condition is called fluctuating light (FL) [1]. During FL, electron flow from photosystem II (PSII) immediately increases upon an abrupt increase in illumination [2]. Concomitantly, CO_2_ fixation needs more time to become fully activated [3], leading to the accumulation of electrons in photosystem I (PSI) electron carriers [4]. If the excess electrons in PSI could not be consumed immediately, reactive oxygen species would generate owing to the electron transfer from reduced P700 to O_2_, causing oxidative damage to the PSI [5,6,7]. The extent of PSI photoinhibition that is induced by FL could be affected by the background low light [7], the intensity of high light [6], and the frequency of low/light cycle [8,9]. Owing to the key role of PSI in the regulation of photosynthesis, PSI photoinhibition strongly suppresses CO_2_ fixation and plant growth [10,11,12,13,14]. Therefore, photoprotection for PSI is fundamental to plant growth under FL [15,16].

PSI over-reduction occurs only when the rate of electron flow from PSII to PSI exceeds the rate of electron sink downstream of PSI [16,17]. Therefore, the PSI redox state can be optimized by donor side regulation and acceptor side regulation [6]. In donor side regulation, the down-regulation of electron flow at either PSII or the cytochrome b6f complex can decrease excitation pressure to PSI and thus alleviate PSI over-reduction [17,18]. In acceptor side regulation, the enhancement of the electron sink downstream of PSI can consume the excess electrons in PSI and thus converts reduced P700 into oxidized P700 [6,19,20]. Flavodiiron proteins (FLVs) consume the excess reducing power in PSI through photo-reduction of O_2_, showing that FLVs protect PSI through acceptor side regulation [16,21]. Meanwhile, the alternative electron flow that is mediated by FLVs is coupled with the formation of ∆pH [16]. As a high ∆pH restricts the plastoquinol oxidation and thus slows down the electron transport at the cytochrome b6f complex [22,23], FLVs might protect PSI through donor side regulation. Some studies have proposed that the functional time of FLVs was shorter than 10 s in the model moss *Physcomitrella patens* [16,24]. However, we recently found that the gymnosperm species *Ginkgo biloba* did not generate a sufficient ∆pH within the initial 10 s after any increase in illumination [25]. Under such conditions, the operation of FLVs-dependent photoreduction of O_2_ was crucial for P700 oxidation and PSI photoprotection. These controversial results require further study to clarify the dynamic change of FLVs activity under FL.

The roles of FLVs in PSI photoprotection have been widely studied in cyanobacteria [26], green algae [27], mosses [16,21,28], and liverworts [29]. However, little is known about the action kinetics of FLVs in ferns and gymnosperms. In *P. patens*, leaves were composed of monolayer cells without stomata. By comparison, stomatal conductance is an important limitation that is imposed to photosynthesis in ferns and gymnosperms [30,31]. We recently found that decreased stomatal conductance led to stronger and prolonged PSI over-reduction under FL in tomato (*Solanum lycopersicum*) and common mulberry (*Morus alba*) [32]. To prevent the risk of PSI over-reduction under FL, the FLVs-dependent electron sink might work longer in ferns and gymnosperms than in *P. patens*.

In angiosperms FLVs are lost during evolution but cyclic electron flow (CEF) around PSI is conserved to sustain photosynthesis under FL [6,15] and other environmental stresses [33,34,35,36,37]. There are two major pathways that are responsible for the operation of CEF, PGR5/PGRL1 and NDH [38,39,40,41,42]. If PGR5/PGRL1-dependent CEF was impaired in angiosperms such as *Arabidopsis thaliana* and rice (*Oryza sativa*), the ∆pH formation would be suppressed under high light, causing PSI over-reduction and thus resulting in severe PSI photoinhibition [6,15,43]. Recent studies indicated that CEF was highly activated upon a transition from low to high light in angiosperms [7,44]. The CEF-dependent ∆pH formation not only slows down the electron flow at the cytochrome b6f complex but also enhances electron downstream of PSI through increasing the ATP/NADPH production ratio [6]. Therefore, CEF strengthens donor side regulation and accelerates acceptor side regulation, both of which alleviate PSI over-reduction under FL. Opposite to angiosperms, the single loss of the PGR5/PGRL1 pathway hardly affected photosynthesis and plant growth under FL in *P. patens* [21], while double losses of PGR5/PGRL1 and NDH pathways strongly accelerated PSI photoinhibition and impaired plant growth [28]. These results indicated that CEF is also indispensable for sustaining photosynthesis and growth under FL in non-angiosperms. However, the dynamic change of CEF under FL and its relationship to ∆pH formation in non-angiosperms have not yet been clarified. In particular, it is unclear how CEF and FLVs interact to regulate photosynthesis under FL.

In addition to photoprotection, CEF regulates the energy balancing in response to a changing environment [45]. Based on the assumption that CO_2_ diffusion conductance is higher in *P. patens* than in ferns and gymnosperms, a relatively lower chloroplast CO_2_ concentration leads to the increase of photorespiration in ferns and gymnosperms. As the energy budget that is required by photorespiration should be balanced by CEF, a relatively lower CEF activity in *P. patens* can satisfy its low capacity of photorespiration. This note was supported by a gradual decrease of CEF activity after an increase in illumination in *P. patens* [24]. By comparison, the high levels of photorespiration in ferns and gymnosperms [31] requires a higher ATP/NADPH production ratio than in *P. patens*. Therefore, we speculate that after the transition from low to high light, the changing patterns of CEF activity in ferns and gymnosperms are different from that in the model moss *P. patens*.

In this study, we measured the dynamic responses of PSI, PSII, and electrochromic shift signals under FL in a fern *Cyrtomium fortune* and a gymnosperm *Nageia nagi*. The aims were: (1) to assess how FLVs interact with ∆pH to regulate PSI redox state under FL; and (2) to examine whether the changing patterns of FLVs and CEF under FL in these two species are different from the model moss *P. patens*. Our results strongly indicate that the time courses of FLVs and CEF in these two species are different from *P. patens* but similar to angiosperms. The specific roles of FLVs and CET under FL are discussed.

## 2. Results

### 2.1. Redox Kinetics of P700 after Transition from Dark to Actinic Light

The redox change kinetics of P700 upon transition to actinic light in dark-adapted leaves is a reliable method to assess the photoreduction of O_2_ that is mediated by FLVs [46,47]. To confirm the existence of FLVs in *Cyrtomium fortunei* and *Nageia nagi*, we first measured the kinetics of P700 redox after transition from dark to actinic light (1809 photons m^−2^ s^−1^) (Figure 1). Both species showed rapid re-oxidation of P700 in 2 s after actinic light was turned on. As the Calvin–Benson cycle was highly inactivated after 60 min dark adaptation, this rapid re-oxidation of P700 was caused by alternative electron downstream of PSI rather than CO_2_ fixation and photorespiration. Furthermore, this rapid re-oxidation of P700 was clearly disappeared when it was measured under anaerobic conditions, which was similar to the phenotype in mutants that were impaired with FLVs. Therefore, photoreduction of O_2_ that was meditated by FLVs contributed to the rapid oxidation of P700 in *C. fortunei* and *N. nagi* during dark-to-light transition.

### 2.2. Changes in PSI and PSII Parameters after Transition from Low to High Light

We next measured the kinetics of PSI and PSII parameters under FL. After transition from 59 to 1809 μmol photons m^−2^ s^−1^, the quantum yield of PSI photochemistry (Y(I)) sharply decreased in both species (Figure 2A). Concomitantly, the P700 oxidation ratio (Y(ND)) rapidly increased to high levels (>0.8) in 10 s (Figure 2B), leading to low PSI acceptor side limitation (Y(NA)) (<0.1) (Figure 2C). Therefore, the PSI over-reduction was obviously prevented in these two studied species after transition from low to high light. Similar to Y(I), the effective quantum yield of PSII photochemistry (Y(II)) sharply decreased by transitioning to high light (Figure 2D). The non-photochemical quenching (NPQ) was rapidly induced and gradually increased to the maximum over time (Figure 2E), resulting in a low quantum yield of non-regulatory energy dissipation in PSII (Y(NO)) (Figure 2F). After the abrupt increase in illumination, the electron transport rate through PSI (ETRI) rapidly increased and was nearly maintained stable (Figure 3A). Concomitantly, the electron transport rate through PSII (ETRII) increased to the peak in approximately 30 s and gradually decreased during the next 90 s (Figure 3B). Therefore, the alternative electron flow that was mediated by FLVs was operational within the initial 20 s after transition from low to high light.

### 2.3. Changes in ΔpH, g_H_^+^, and CEF after Transition from Low to High Light

As the proton gradient (ΔpH) across the thylakoid membranes and the activity of thylakoid proton conductance (*g*_H_^+^) play important roles in photosynthetic regulation under excess light energy, we measured the ECS signals to estimate the kinetics of ΔpH and *g*_H_^+^ after transition from low to high light. We found that in *C. fortunei* and *N. nagi*, the values of ΔpH after this light transition for 20 s were significantly lower than those after this light transition for 60 s and 120 s (Figure 4). These results indicated that *C. fortunei* and *N. nagi* could not generate a sufficient ΔpH at least within the initial 20 s after the light intensity increased abruptly. Such insufficient ΔpH might be related to the relatively high *g*_H_^+^ (Figure 5). After transition from low to high light, *g*_H_^+^ first decreased and finally re-increased (Figure 5). Such a re-increase of *g*_H_^+^ suggested that the Calvin–Benson cycle was re-activated. To evaluate the kinetics of CEF under FL, we calculated time courses of the relative proton flux through the thylakoid membrane (*v*_H_^+^) and the ratio of *v*_H_^+^/ETRII. After transition from low to high light, *v*_H_^+^ and *v*_H_^+^/ETRII ratio rapidly increased to the peak in 10 s in both species, followed by the rapid decreases in 20 s and the subsequent re-increase in 120 s (Figure 6). These results indicated that CEF was highly stimulated within the first 10 s but rapidly decreased in the subsequent seconds. After the Calvin cycle was highly activated under high light, CEF was re-activated to sustain photosynthesis.

To examine the role of ΔpH in photosynthetic regulation under FL, we plotted the values of ΔpH, NPQ, and Y(ND) after transition from low to high light. Positive relationships between ΔpH and NPQ were found in both species (Figure 7A), indicating that the gradual increase of ΔpH induced NPQ and thus protected PSII against excess light energy. However, the dynamic changes of ΔpH hardly affected the values of Y(ND) in both species (Figure 7B), pointing out that in these two non-angiosperms the PSI redox state under FL was not controlled by ΔpH. Therefore, the dynamic formation of ΔpH under FL had different effects on photoprotection for PSI and PSII in non-angiosperms.

## 3. Discussion

The FLVs-mediated alternative electron sink is operational in photosynthetic organisms except in angiosperms [24,48]. Recent studies have documented the critical roles of FLVs in cyanobacteria, green algae, mosses, and liverworts under FL [16,26,27,29]. In these groups, *flv* mutants showed severe PSI photoinhibition and stunted plant growth if they are exposed to FL [21,28]. However, little is known about the role of FLVs in photosynthetic regulation under FL in the residual other two groups of non-angiosperms, ferns and gymnosperms. Here, we document that FLVs-mediated photoreduction of O_2_ contributes to a rapid oxidation of P700 after any increase in light intensity in *Cyrtomium fortune* (fern) and *Nageia nagi* (gymnosperm) (Figure 1 and Figure 2). Furthermore, we found that the action kinetics of FLVs activity were largely correlated with the ∆pH formation. In addition, the role CEF in photosynthetic regulation under FL in them was energy balancing rather than photoprotection.

Although FLV activity is proven to be the one of most important regulators in non-angiosperms when exposed to FL, the specific protection mechanisms have not yet been well clarified. Generally, alternative electron transport that is mediated by FLVs not only enhances electron downstream of PSI but also contributes to the ∆pH formation across the thylakoid membranes. Therefore, some scholars propose that the ∆pH-dependent photosynthetic control at the cytochrome b6f complex is an important mechanism behind how FLVs oxidize P700 under FL [16]. However, this scheme is now challenged by a recent study on the gymnosperm *Ginkgo biloba* [25]. After transition from dark or low light to high light, *G. biloba* did not build up a sufficient ∆pH in 10 s but P700 was highly oxidized, leading to a hypothesis that the ∆pH-dependent photosynthetic control played a minor role in FLVs-mediated P700 oxidation [25]. In the model moss *P. patens*, FLVs-dependent alternative flow was functional in the first seconds up to 10 s but was undetectable during prolonged illumination [16,24]. If the FLV activity just functioned within the first 10 s after light abruptly increased, ETRII would get the first peak within the first 10 s. When FLV activity is inactivated, ETRII will gradually increase owing to the activation of the Calvin cycle [24]. However, we observed that ETRII peaked after transition from low to high light for 30 s in *C. fortune* and *N. nagi*, followed by the subsequent decrease in 120 s (Figure 3B). This result indicated that the action time of FLVs in them lasted at least 30 s. Therefore, the function time of FLVs is likely different among different groups of non-angiosperms.

A question arises why FLVs work longer in *C. fortune* and *N. nagi* than in *P. patens*. One possible explanation is that the work time of FLVs is influenced by the kinetics of ∆pH formation. Many previous studies on angiosperms have documented that ∆pH is the key signal determining the redox state of PSI under high light and FL [38]. For example, if the ∆pH formation was suppressed by the impairment of CEF or enhanced the activity of chloroplast ATP synthase, PSI over-reduction and severe PSI photoinhibition would be observed under FL [49,50]. However, these lethal phenotypes can be rescued by the introduction of FLVs [6]. Furthermore, introduction of FLVs into WT plants of *Arabidopsis thaliana* and rice can prevent PSI over-reduction under FL but do not affect the steady-state photosynthesis [4,51]. Therefore, FLVs activity is particularly seminal when the ∆pH is low but is dispensable under conditions of high ∆pH. In the present study, we found that *C. fortune* and *N. nagi* needed approximately 30 s to accomplish the buildup of ∆pH (Figure 4), which was consistent with the duration time of FLVs activity (Figure 3B). After transition to high light for 120 s, ∆pH was formed sufficiently in *C. fortune* and *N. nagi* (Figure 4). Concomitantly, FLVs was inactivated as indicated by the relatively low ETRII values (Figure 3B). Therefore, after transition from low to high light, FLVs and ∆pH alternately optimize the PSI redox state in non-angiosperms.

Compared with FLVs, CEF plays a weaker role in photoprotection for PSI under FL in the model moss *P. patens* [21]. In *P. patens*, single impairment of the PGR5/PGRL1 pathway hardly affected the PSI activity and plant growth under FL [21], while double impairment of the PGR5/PGRL1 and NDH pathways largely reduced biomass production [28]. Here, we found that in *C. fortune* and *N. nagi,* CEF was stimulated in the first 10 s after an abrupt increase in illumination (Figure 6B), which was accompanied by an insufficient ∆pH (Figure 4). Therefore, CEF probably helped the rapid formation of ∆pH under FL in them, which was similar to the phenotypes in angiosperms. Such CEF-dependent ∆pH formation favored the induction of NPQ and thus contributed to photoprotection for PSII (Figure 7A). However, unlike the importance of CEF in regulating the PSI redox state under FL in angiosperms, the PSI redox state was not influenced by ∆pH because of the weak correlation between P700 oxidation and ∆pH (Figure 7B). Therefore, in the presence of FLVs, the rapid oxidation of PSI in *C. fortune* and *N. nagi* was independent of the transient stimulation of CEF.

After transition to high light for 120 s, the relatively stable ETRII was accompanied with the re-increase of CEF activity (Figure 6B). This re-increase of CEF activity contributed to the maintenance of ∆pH under constant high light, which balanced the energy budget for primary metabolism. The CEF activity can be regulated by adenylate status. Generally, CEF is activated when the stromal ATP level is low, but it is downregulated when the ATP level increases [52]. Within the first 10 s after transition from low to high light, the low stroma ATP level initiated the transient stimulation of CEF. Subsequently, the operation of FLVs activity increased the stroma ATP level and thus slowed down the CEF activity. Under steady state photosynthesis, the full activation of CO_2_ assimilation and the inactivation of FLVs activity decreased the stroma ATP level, which induced the re-activation of CEF to maintain normal adenylate status. Therefore, the dynamic change of CEF activity under FL further explains why CEF is essential for sustaining photosynthesis in non-angiosperms.

In summary, we examined the kinetics of FLVs activity and CEF after transition from low to high light in the fern *C. fortune* and the gymnosperm *N. nagi*. We found that within the first 30 s after light abruptly increased, FLVs were functional to avoid PSI over-reduction under conditions of insufficient ∆pH. The functional time of FLVs in *C. fortune* and *N. nagi* was much longer than those in the model moss *P. patens*. CEF was strongly activated within the first 10 s after the light abruptly increased, which played a major role in energy balancing rather than the regulation of the PSI redox state. When FLVs were inactivated during steady-state photosynthesis, CEF was re-activated to favor photoprotection and sustain photosynthesis. These results provide new insights into how FLVs and CEF interact to regulate photosynthesis in non-angiosperms.

## 4. Materials and Methods

### 4.1. Plant Materials

A fern species *Cyrtomium fortunei* J. Sm. (Dryopteridaceae) and a gymnosperm species *Nageia nagi* (Thunberg) Kuntze (Podocarpaceae) were chosen in this study. The plants were grown in a greenhouse at the Kunming Botanical Garden, Yunnan, China (102°44′31″ E, 25°08′24″ N, 1950 m of elevation). The light conditions were 40% full sunlight (the maximum value was 800 μmol photons m^−2^ s^−1^), day/night air temperatures 30/20 °C, and a relative humidity of approximately 60–70%. The plants were grown in plastic pots. To prevent any water and nutrient stress, the plants were fertilized and watered regularly. Photosynthetic measurements were conducted on mature but not senescent leaves.

### 4.2. P700 Redox Kinetics Measurements

The redox kinetics of P700 was measured using a Dual-PAM 100 measuring system (Heinz Walz, Effeltrich, Germany) by illumination on dark-adapted leaves. After the inactivation of the Calvin–Benson cycle by dark adaptation for at least 60 min, intact leaves were illuminated at 1809 μmol photons m^−2^ s^−1^ for 5 s under atmospheric air conditions at approximately 25 °C [46].

### 4.3. P700 and Chlorophyll Fluorescence Measurements

P700 and chlorophyll fluorescence were measured simultaneously at 25 °C using a Dual-PAM 100 measuring system (Heinz Walz, Effeltrich, Germany). A saturating pulse (20,000 μmol photons m^−2^ s^−1^, 300 ms) was used to measure the maximum fluorescence intensity (*F_m_*) and the maximum photo-oxidizable P700 (*P_m_*) after dark adaptation for at least 15 min. Subsequently, 15 min illumination at 759 μmol photons m^−2^ s^−1^ was conducted to activate photosynthetic electron sinks, and then light intensity was changed to 59 μmol photons m^−2^ s^−1^ for 5 min. Afterwards, light intensity abruptly increased to 1809 μmol photons m^−2^ s^−1^. During the follow-up 2 min, P700 and chlorophyll fluorescence parameters were recorded.

The chlorophyll fluorescence parameters were calculated as follows: the quantum yield of PSII photochemistry, Y(II) = (*F_m_’* − *F_s_*)/*F_m_’*; non-photochemical quenching in PSII, NPQ = (*F_m_* − *F_m_’*)/*F_m_’*; the quantum yield of non-regulatory energy dissipation in PSII, Y(NO) = *F_s_*/*F_m_*. *F_m_* and *F_m_’* were the maximum fluorescence intensity and were recorded after dark and light acclimation, respectively. *F_s_* is the pre-trigger fluorescence intensity. The PSI parameters were calculated as follows: the quantum yield of PSI photochemistry, Y(I) = (*P_m_’* − *P*)/*P_m_*; the quantum yield of PSI non-photochemical energy dissipation due to donor side limitation, Y(ND) = *P*/*P_m_*; the quantum yield of PSI non-photochemical energy dissipation due to acceptor side limitation, Y(NA) = (*P_m_* − *P_m_’*)/*P_m_*. The photosynthetic electron transport rate was calculated as ETRI (or ETRII) = PPFD × Y(I) [or Y(II)] × 0.84 × 0.5, light absorption is assumed to be 0.84 of the incident irradiance, and 0.5 is the fraction of absorbed light reaching PSI or PSII.

### 4.4. Electrochromic Shift (ECS) Analysis

The ECS signal was monitored using a Dual PAM-100 that was equipped with a P515/535 emitter-detector module by recording the change in absorbance at 515 nm [53]. Before the ECS measurement, a single turnover flash (ECS_ST_) was measured after dark-adaptation for at least 60 min [54]. Subsequently, photosynthetic induction was conducted at 759 μmol photons m^−2^ s^−1^ for 15 min. Afterward, the leaves were illuminated at 59 μmol photons m^−2^ s^−1^ for 5 min, and then the ECS signal was recorded after transition to 1809 μmol photons m^−2^ s^−1^ for 10 s. Subsequently, the leaves were repeatedly acclimated to 59 μmol photons m^−2^ s^−1^ for 5 min, and then the ECS signal was measured after transition to 1809 μmol photons m^−2^ s^−1^ for 20 s. Similar ECS signals were measured after transition from 59 to 1809 μmol photons m^−2^ s^−1^ for 30 s, 40 s, 50 s, 60 s, and 120 s. The ECS dark interval relaxation kinetics (DIRK_ECS_) were used to calculate the activity of chloroplast ATP synthase (*g*_H_^+^) and proton gradient (ΔpH) across the thylakoid membranes [55,56]. All the ΔpH levels were normalized against the magnitude of ECS_ST_. The relative proton flux through the thylakoid membrane (*v*_H_^+^) in the light was calculated from the maximal drop in the ECS signal (ECS_t_) during a 300-ms DIRK_ECS_ and the value of *g*_H_^+^: *v*_H_^+^ = ECS_t_ × *g*_H_^+^ [56,57]. The relative CEF activation state was estimated by the *v*_H_^+^/ETRII ratio.

### 4.5. Statistical Analysis

Paired *t*-tests were used to determine whether significant differences in *g*_H_^+^ and ΔpH existed between the different treatments (*α* = 0.05). The software SigmaPlot 10.0 was used for graphing and fitting.

## Figures and Tables

**Figure 1 cells-11-02768-f001:**
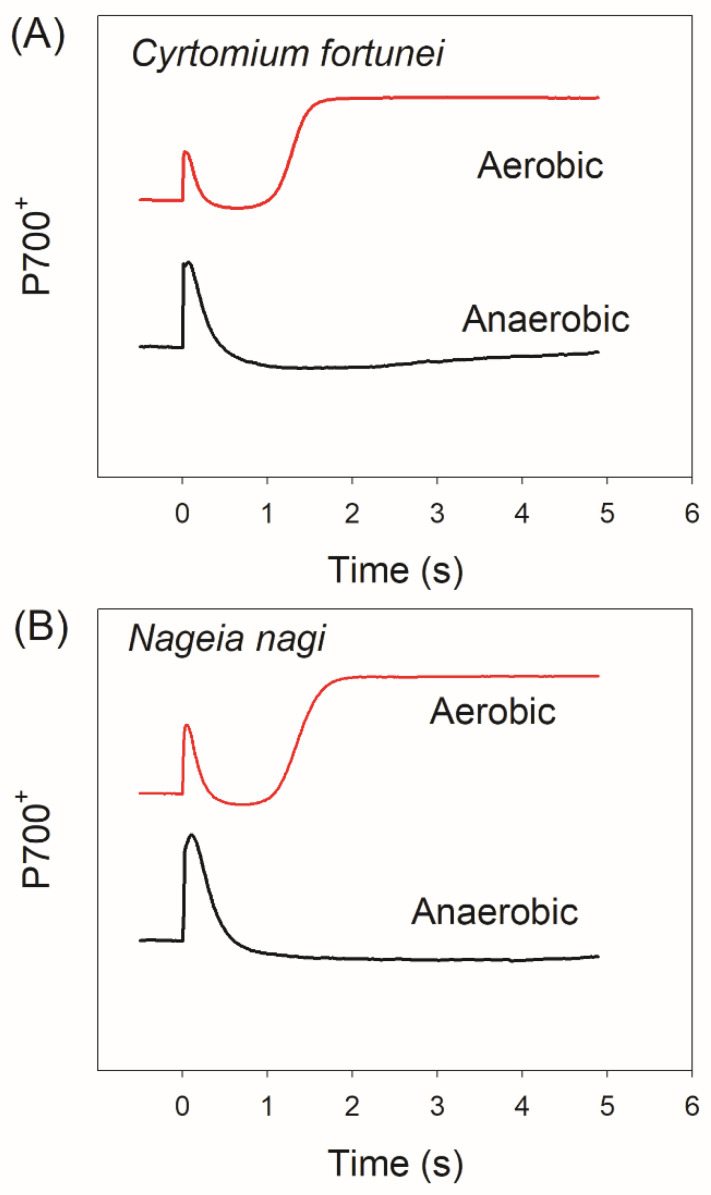
Redox kinetics of P700 upon transition from dark to actinic light (1809 μmol photons m^−2^ s^−1^) in leaves of *Cyrtomium fortune* (**A**) and *Nageia nagi* (**B**) measured under aerobic and anaerobic conditions. The data are the means of five independent leaves from five independent plants.

**Figure 2 cells-11-02768-f002:**
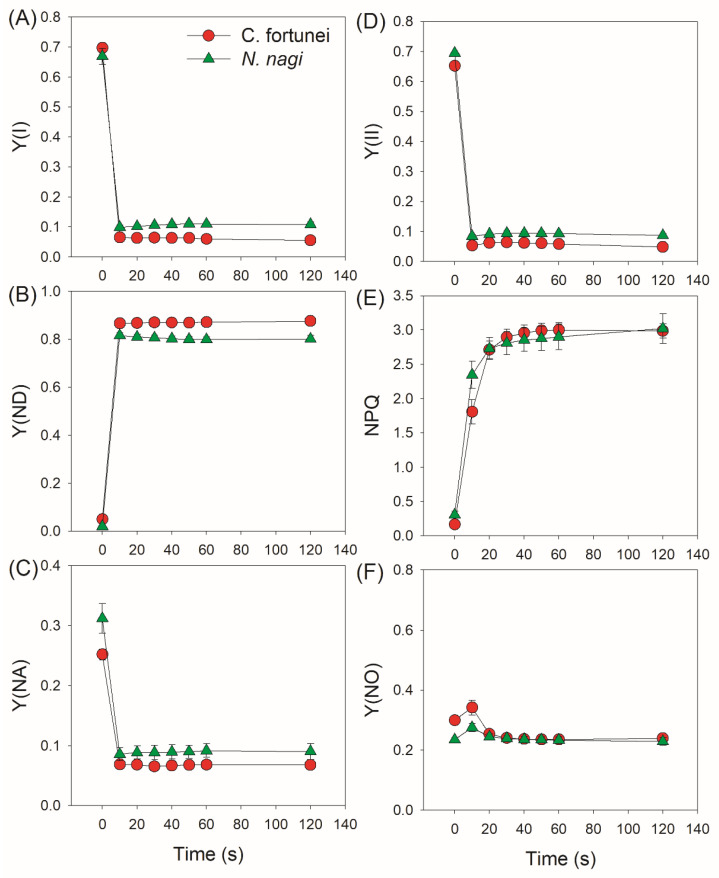
Changes in the quantum yields of PSI and PSII after transition from 59 (t = 0) to 1809 μmol photons m^−2^ s^−1^ in leaves of *Cyrtomium fortune* and *Nageia nagi*. (**A**) Y(I) represents the quantum yield of PSI photochemistry; (**B**) Y(ND), the quantum yield of PSI non-photochemical energy dissipation due to donor side limitation; (**C**) Y(NA), the quantum yield of PSI non-photochemical energy dissipation due to acceptor side limitation; (**D**) Y(II) represents the quantum yield of PSII photochemistry; (**E**) NPQ, non-photochemical quenching in PSII; (**F**) Y(NO), the quantum yield of PSII non-regulatory energy dissipation. The data are the means ± SE (n = 5).

**Figure 3 cells-11-02768-f003:**
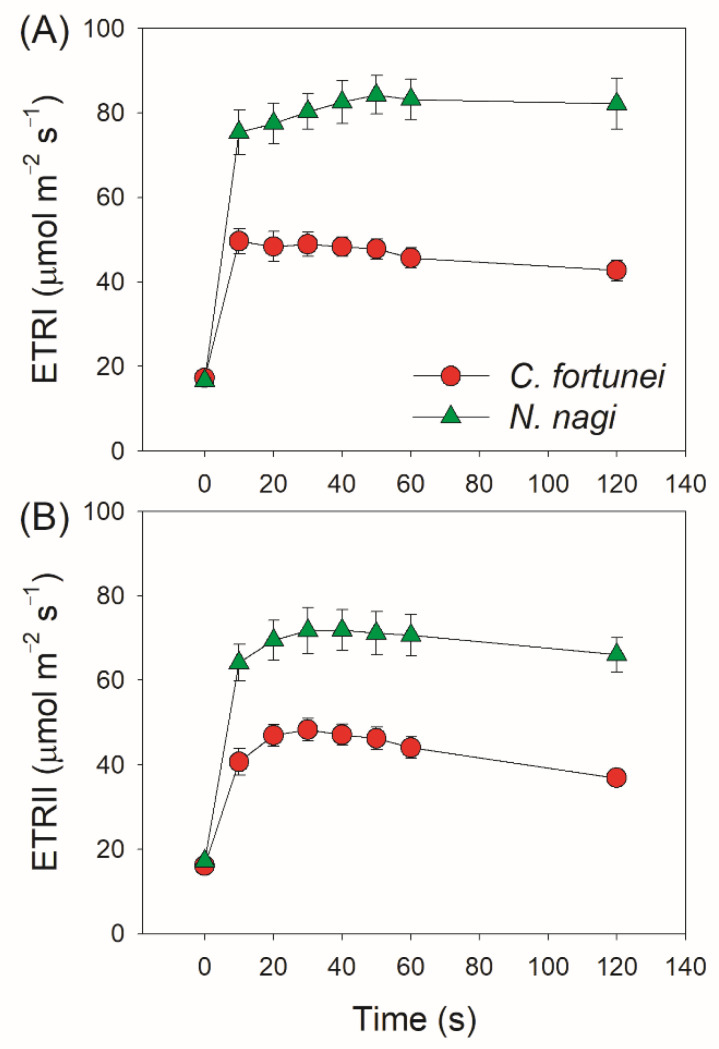
Changes in the photosynthetic electron transport rates after transition from 59 (t = 0) to 1809 μmol photons m^−2^ s^−1^ in leaves of *Cyrtomium fortune* and *Nageia nagi*. (**A**) ETRI represents electron transport rate through PSI; (**B**) ETRII, electron transport rate through PSII. The data are the means ± SE (n = 5).

**Figure 4 cells-11-02768-f004:**
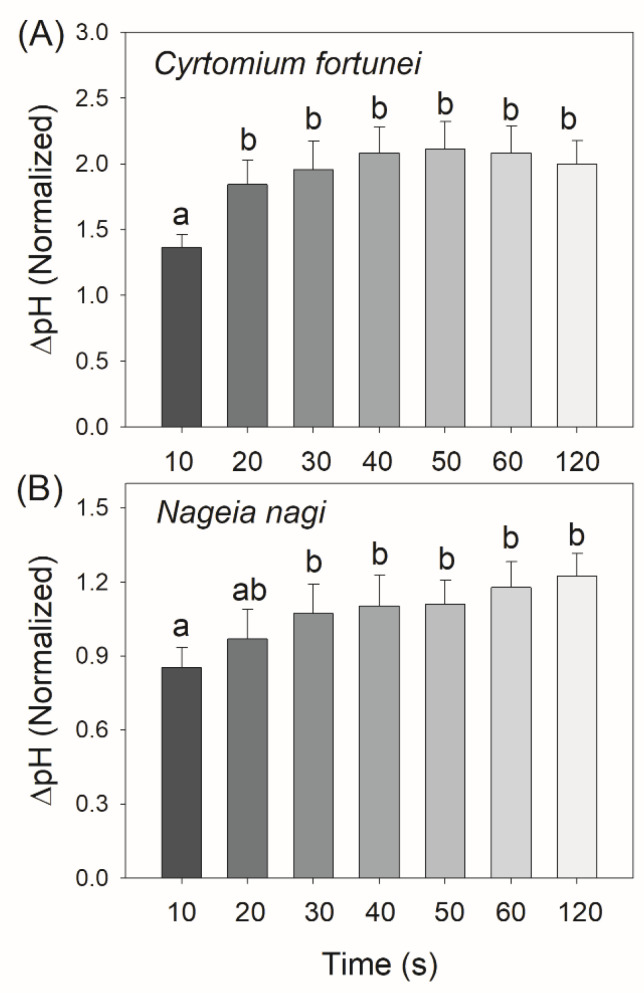
Changes in the trans-thylakoid proton gradient (∆pH) after transition from 59 to 1809 μmol photons m^−2^ s^−1^ in leaves of *Cyrtomium fortune* (**A**) and *Nageia nagi* (**B**). All the ΔpH levels were normalized against the magnitude of ECS_ST_. The data are the means ± SE (n = 5). Different letters indicate significant differences between the different treatments.

**Figure 5 cells-11-02768-f005:**
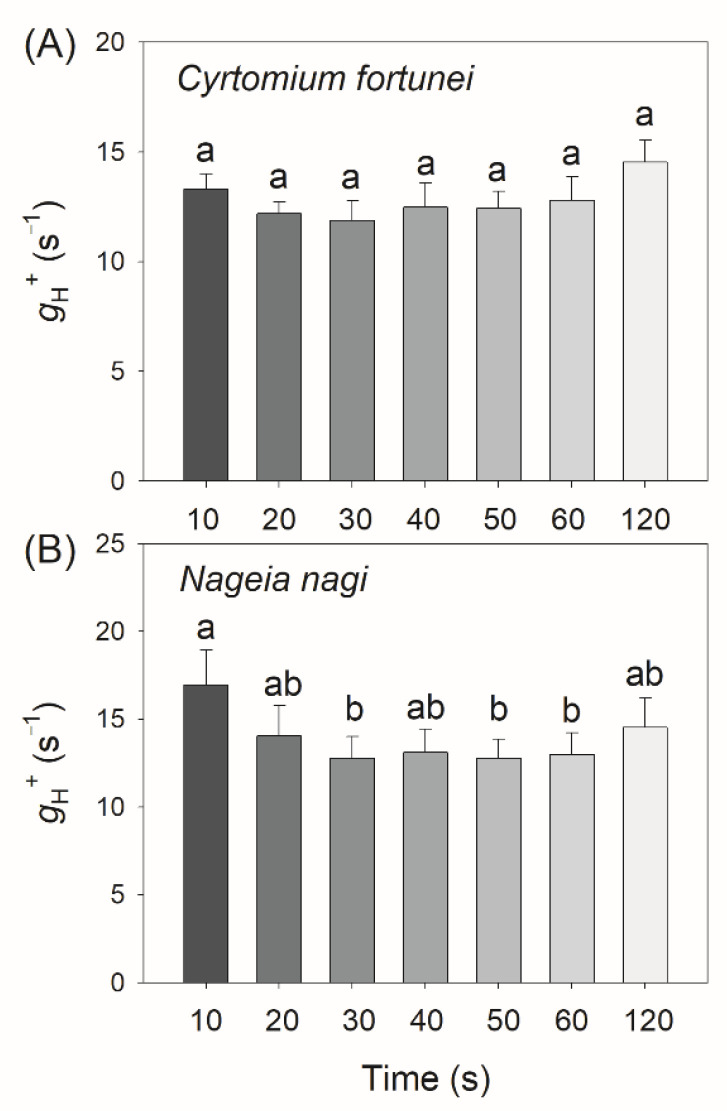
Changes in the activity of chloroplast ATP synthase (*g*_H_^+^) after transition from 59 to 1809 μmol photons m^−2^ s^−1^ in leaves of *Cyrtomium fortune* (**A**) and *Nageia nagi* (**B**). The data are the means ± SE (n = 5). Different letters indicate significant differences between the different treatments.

**Figure 6 cells-11-02768-f006:**
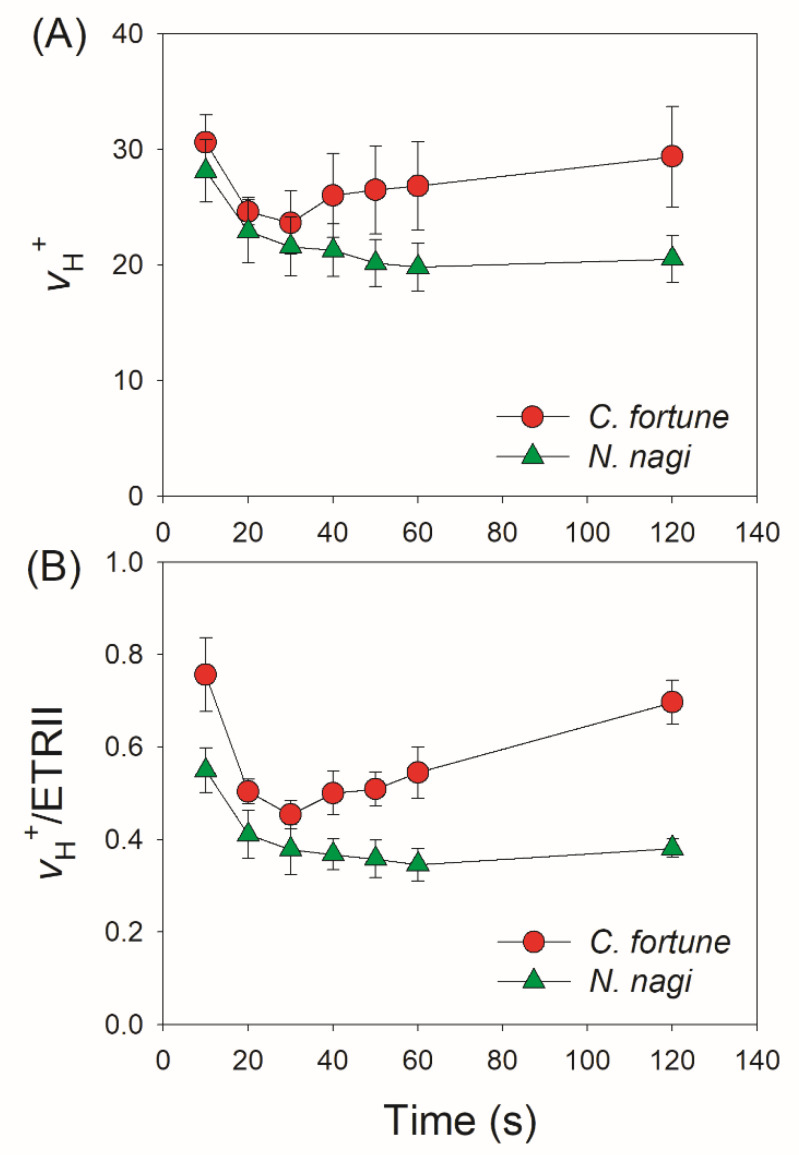
Changes in the relative proton flux through the thylakoid membrane (*v*_H_^+^) (**A**) and relative CEF (**B**) after transition from transition from 59 to 1809 μmol photons m^−2^ s^−1^ in leaves of *Cyrtomium fortune* and *Nageia nagi*. The relative CEF was estimated by the *v*_H_^+^/ETRII ratio. The data are the means ± SE (n = 5).

**Figure 7 cells-11-02768-f007:**
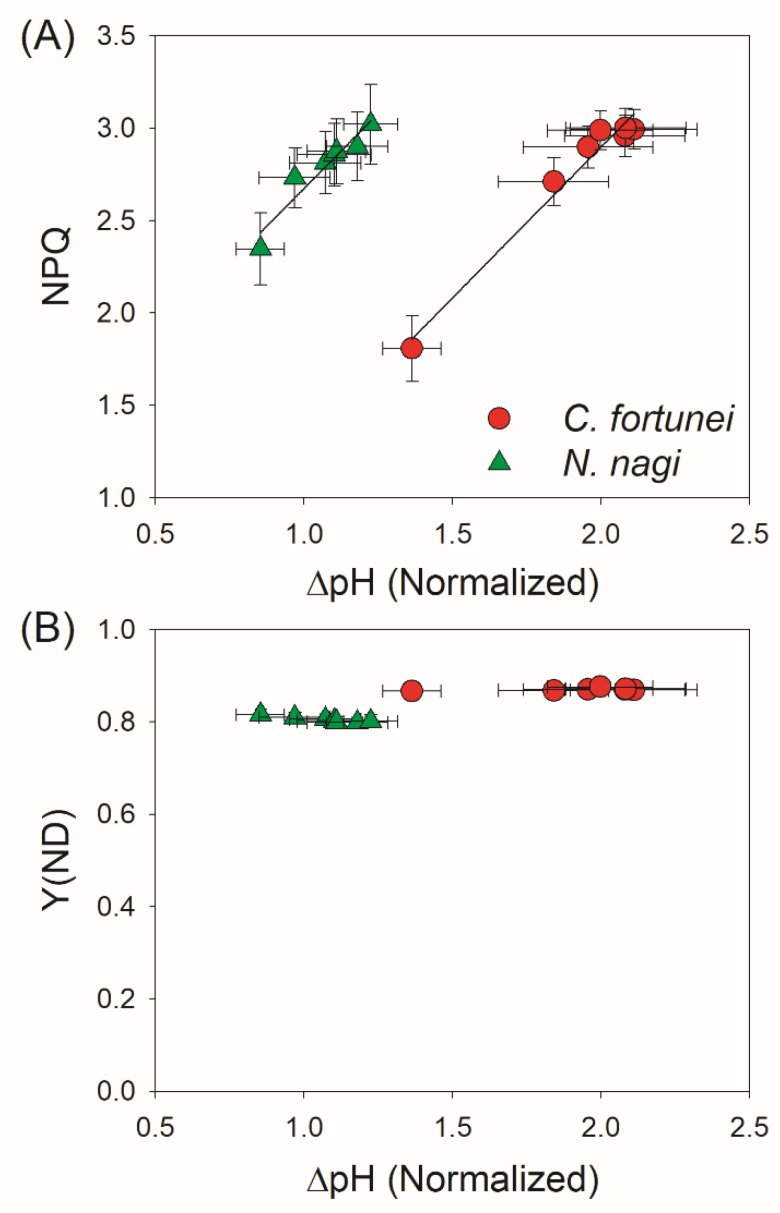
Changes in NPQ (**A**) and Y(ND) (**B**) as a function of ∆pH after transition from 59 to 1809 μmol photons m^−2^ s^−1^ in leaves of *Cyrtomium fortune* and *Nageia nagi*. The data are the means ± SE (n = 5).

## Data Availability

The data presented in this study are available on request from the corresponding author.

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
