# Peer review of "The Dynamic Changes of Alternative Electron Flows upon Transition from Low to High Light in the Fern Cyrtomium fortune and the Gymnosperm Nageia nagi"

_cells, 2022, doi:10.3390/cells11172768_

Round 1

Reviewer 1 Report

The present work is very interesting and authors explained very good way about their findings. But I have some concern about the manuscript. The following points should be needed address before acceptance of this article.

-The format of the manuscript is not properly followed.

-The authors should carefully review the typography and correct the grammatical errors in the whole manuscript.

Abstract: The abstract is OK.

Keywords: The keywords are appropriate but should be arranged alphabetically.

Introduction

The introduction needs rigorous modification. The author write the introduction as a descriptive form and its too long but it should be precisely revised the introduction. There should be needs a link up among the whole introduction. The author should cited recent references related to this study.

10.1038/s41438-021-00460-y

10.1007/s11120-019-00631-y

10.1016/j.plaphy.2021.08.002

Materials and Methods

The methodology is written very well but author should try to avoid grammartical errors.

Results

The author explained the each parameters profoundly but my suggestions the quality of the figures are not up to mark. Another, it would be better if author merged the figure 2 and 3 together for comparison easily. In figure 6 missing lettering.

Discussion : OK.

Conclusions

The conclusion is well summarized but not in a point of order. Write sequentially.

References

References are not properly formatted. Kindly follow the Instructions for Authors.

Author Response

Thanks a lot for these important comments.

  1. We have revised the manuscript according to the Instructions for Authors.
  2. We have corrected the grammatical errors in the whole text.
  3. The keywords have been arranged alphabetically.
  4. The introduction has been precisely revised and the recommended references related to this study have been added.
  5. Figures have been revised according to the reviewer’s suggestions. 

Reviewer 2 Report

The paper by Cheng et al deals with changes in alternative electron flow after a transition from low light to high light. The topic is very interestìsting and it it may be suited for pubblication in Cells.  However I have a couple of concerns:

i) - The title in my opinion does not correctly describe the content of the work: usually, the term 'fluctuating light' is referred to a plant grown under a given light to which a second, higher light intensity, is periodically superimposed for a long period of time (days); here, just a change from low light to high light is used, and so it cannot be called fluctuating ligfht;

ii) - although it may have some meaning to check the presence of Flv by the method reported in the first paragraph of the Result section, as the species used in this paper has not been previously characterized, a immunoblot where the presence (and amount, at least relative) of flavodiiron proteins is directly demonstrated, should be reported. 

iii) - Figures 2,3,4: does t=0 represent the transition from low to high light? If yes, it should be make clearer  

iv) - in my opinion, the Discussion section could be shortened

Author Response

Thanks a lot for these important comments.

  1. We have replaced “under fluctuating light” by “upon transition from low to high light”.
  2. Indeed, an immunoblot is very useful for confirming flavodiiron proteins in these two studied species. To detect the expression of flavodiiron proteins by western blot, the corresponding antibody for these two species is needed. However, we try our best and cannot buy the specific antibodies after because these two studied species are not model non-angiosperms. Alternatively, the kinetics of P700 redox upon transition from dark to actinic light has been proved to be a reliable method, which was used to monitor the presence of Flvs in these two studied species.

  3. Indeed, in Figures 2, 3 and 4, t = 0 represents the transition from low to high light. We have added this in these figure legends.
  4. We have shortened the Discussion section to avoid redundancy. 

Round 2

Reviewer 2 Report

the manuscript has been sufficiently improved to be accepted in the prsesnt form